# DSE-NN: Deeply Supervised Efficient Neural Network for Real-Time Remote Photoplethysmography

**DOI:** 10.3390/bioengineering10121428

**Published:** 2023-12-15

**Authors:** Seongbeen Lee, Minseon Lee, Joo Yong Sim

**Affiliations:** Department of Mechanical Systems Engineering, Sookmyung Women’s University, Seoul 04310, Republic of Korea; seongbeen@sookmyung.ac.kr (S.L.); todaydream77@sookmyung.ac.kr (M.L.)

**Keywords:** deep supervision, light-weight, remote photoplethysmography

## Abstract

Non-contact remote photoplethysmography can be used in a variety of medical and healthcare fields by measuring vital signs continuously and unobtrusively. Recently, end-to-end deep learning methods have been proposed to replace the existing handcrafted features. However, since the existing deep learning methods are known as black box models, the problem of interpretability has been raised, and the same problem exists in the remote photoplethysmography (rPPG) network. In this study, we propose a method to visualize temporal and spectral representations for hidden layers, deeply supervise the spectral representation of intermediate layers through the depth of networks and optimize it for a lightweight model. The optimized network improves performance and enables fast training and inference times. The proposed spectral deep supervision helps to achieve not only high performance but also fast convergence speed through the regularization of the intermediate layers. The effect of the proposed methods was confirmed through a thorough ablation study on public datasets. As a result, similar or outperforming results were obtained in comparison to state-of-the-art models. In particular, our model achieved an RMSE of 1 bpm on the PURE dataset, demonstrating its high accuracy. Moreover, it excelled on the V4V dataset with an impressive RMSE of 6.65 bpm, outperforming other methods. We observe that our model began converging from the very first epoch, a significant improvement over other models in terms of learning efficiency. Our approach is expected to be generally applicable to models that learn spectral domain information as well as to the applications of regression that require the representations of periodicity.

## 1. Introduction

Heart pulse is an important physiological signal that reflects physical and emotional activities. Monitoring heart pulse can be used for various applications such as arrhythmia detection in human-robot interaction [1], cardiology [2], emotion recognition [3], thermal comfort management [4], mental healthcare [5], and exercise training support [6]. Remote photoplethysmography (rPPG) [7] has been introduced in recent years, to measure heart pulse without any contact with sensors [8,9] to overcome the limitations of optoelectronic sensors, which can cause discomfort and limit long-term monitoring.

The on-contact rPPG evaluates back-scattered light from skin remotely using RGB cameras. In earlier studies of rPPG [10,11], the analysis was carried out through face detection, region of interest (ROI) selection, rPPG signal recovery, and signal post-processing. Studies have been conducted to replace these feature-based rPPG methods with deep learning-based methods. As deep learning-based approaches, 2D CNN, 2D CNN with LSTM, and 3D CNN-based methods have been proposed for processing spatiotemporal information. The 3D CNN contributed to improving the accuracy by allowing the neural network to learn feature extraction for temporal information by performing convolution in the spatiotemporal domain. However, in comparison to 2D CNN, these 3D CNN methods have limitations in that the number of parameters increases and the operation speed becomes slow. Furthermore, as the CNN layers get deeper, the optimization becomes challenging due to gradient vanishing and initialization problems.

To alleviate these difficulties, researchers introduced regularization methods based on residual connections as well as auxiliary tasks for skin segmentation and average HR (heart rate) regression along with the PPG prediction. Yin et al. [12] employed residual blocks of spatiotemporal networks that can help learn and optimize neural networks even with deep layers. To deal with the problem of heavy computation and large complexity in 3D CNN, the following studies have proposed computationally efficient rPPG models. Gudi et al. [13] obtained a speed of 30 frames per second by combining the skin pixel selection [14] and plane-orthogonal to skin (POS) methods [15]. Liu et al. [16] proposed temporal shift convolutional attention network to achieve efficient temporal modeling and reduce on-device latency, resulting in a processing speed of 150 frames per second. To improve the efficiency of 3D CNNs for rPPG, Kuang et al. [17] introduced 3D depth-wise separable convolution. Recently, Comas et al. [18] reported a lightweight neural model by using convolutional derivatives and time-shift invariant loss. Botina-Monsalve et al. [19] suggested an ultra-light 3D CNN rPPG model by decreasing the input size with a frequency-based loss function. However, the concepts from these earlier studies are difficult to readily adapt to rapidly evolving new deep neural networks, and these neural networks, being black box models, are deficient in explaining why and how these proposed methods work well for rPPG.

To address these limitations, we introduce a deeply supervised efficient neural network (DSE-NN) based on deep supervision, network visualization, and hyperparameter optimization. Progress in the field of deep neural networks has been accelerated by the development of better tools for visualizing and interpreting neural nets. However, understanding of learned features of the spatiotemporal neural network, especially what computations rPPG deep learning models perform at intermediate layers, has lagged behind. In our study, inspiration for deep supervision and reduction of model complexity came from the visualization of the intermediate layers of PhysNet [20]. PhysNet has been widely used as a baseline in many previous studies without concern for a large number of channels in the intermediate layers. Through the visualization of these intermediate layers, we observed the periodic pattern that can be correlated with ground truth PPG signals in the activation maps of intermediate layers. On this basis, DSE-NN adopted deep supervision and channel reduction strategies to facilitate the convergence of the model and increase the accuracy (Figure 1). As far as we know, this is the first study that utilizes deep supervision for rPPG prediction, and our proposed approach can be applied generally to other deep learning approaches.

The main contribution of this paper is as follows:We introduce spectral deep supervision to rPPG models, which learns quickly through spectral loss of intermediate layers that facilitates convergence and increases accuracy.Through investigation of visualization of intermediate layers we provide intuitions about the need for efficient networks as well as their guided supervision of the hidden neurons.Using the obtained basis for reducing model complexity, we optimized the spatiotemporal network for weight reduction accompanied by ablation studies, resulting in the number of trainable parameters being reduced by a factor of 11.

We present an overview of our model’s architecture in Figure 1. Our approach is based on the PhysNet [20] encoder-decoder model. We propose a lightweight deep supervision network designed for rPPG. The input of the proposed model is a video sequence, and the output is a predicted rPPG signal. The baseline model is represented by gray boxes, while the enhancements constituting our proposed network are indicated in blue. The size of these boxes reflects the lightweight nature of the respective network components, with smaller boxes denoting a more compact and efficient network structure. Furthermore, we employ spectral loss for deep supervision, utilizing outputs from intermediate layers, which is a key aspect of our proposed enhancements. This approach ensures precise signal extraction while maintaining a streamlined network architecture.

## 2. Related Works

### 2.1. rPPG Based on Deep Learning

Changes in light absorption for blood volume pulses are not only too small to be visible to the human eye, they can also be sensitively changed by changes in light and noise introduced by head movements. To solve this problem, many approaches based on color space projection and statistical signal decomposition have been proposed, however, these methods are unable to take into account non-linearity and separate convolved noise signals. Additionally, the signal processing-based methods cause spatial information loss due to averaging pixels in ROIs, which are chosen in a fixed or random. Researchers have started using neural networks to recover PPGI signals and have achieved better experimental results than conventional methods [9]. Since deep learning is a data-driven method, the training set contains samples of different scenarios, ensuring robustness and flexibility in real-world applications. Unlike static images, rPPG reconstructions must consider both spatial and temporal characteristics. Spatial features include skin texture, edge, and background information, while temporal features describe changes in spatial features over time, including features such as respiration, heart rate, and tremor. For these problems, various methods of 2D CNN [21], 3D CNN and CNN with RNN (Convolutional Recurrent Neural Network) [22] have been used to estimate rPPG and HR. DeepPhys [21] reported a convolutional attention network consisting of a motion model and an appearance model. To use time information, normalized differences in two consecutive frames were used as input to the motion model. Also, the appearance model is given the original frame as input and acts as an attention module. The network learns spatial masks simultaneously for the proper region of interest detection, thereby recovering rPPG and HR signals. Since rPPG is estimated from video data, using 2D CNN requires additional processes to consider the temporal characteristics of the video. PhysNet [20] proposed a 3D CNN model that can perform rPPG measurements while simultaneously analyzing the spatial and temporal characteristics of images. Recently, Yin et al. [12] proposed (2 + 1) D convolution for rPPG to separate time and spatial information for the robustness of the model. Unlike other 3D CNN models, the authors applied convolution separately in time and space, and stacked them as a block to downstream networks to estimate the final ppg signal. Models using 3D CNN are effective at analyzing spatiotemporal characteristics, but due to their large number of parameters, they require more computation and longer interference time than other models.

### 2.2. Efficient Neural Networks for rPPG

Recent rPPG models have proposed methods to solve the problems of computation cost and long training and inference time. For example, Gudi et al. [13] reported a method to increase the efficiency of the rPPG system by extracting the face region using an independently developed deep learning framework and solving the rPPG extraction problem through a simplified signal processing method. EVM-CNN [23] proposed a model that reduces the computational burden by predicting the heart rate using a neural network after performing Eulerian Video Magnification on the region of interest extracted as above. RTrPPG [19] proposed a method to increase the efficiency of 3D CNN by compressively reducing the spatial information of the input image from the beginning by optimizing it through the ablation study of the existing 3D CNN model. Kuang et al. [17] used 3D depth-wise separable convolution and a structure based on mobilenet v3 combined with a lightweight attention block to reduce the time complexity of the network.

### 2.3. Deep Supervision

GoogLeNet [24] introduced deep supervision with auxiliary loss in that it includes two additional auxiliary classifiers to train the 22-layer deep inception architecture. As the neural network gets deeper, the vanishing gradient problem worsened, making it difficult to converge, which was solved by the auxiliary layers that strengthen the gradient in the intermediate layers and encourage discrimination in the lower stages of the network. During backpropagation, a weight of 0.3 was used for the auxiliary classifiers to prevent it from hindering the learning of the primary classifier that was used alone for inference. Deeply supervised nets [25] further solidified the idea of deep supervision that provides integrated direct supervision to the hidden layers, rather than providing supervision only at the output layer and propagating this supervision back to earlier layers. These deeply supervised objective functions were shown not only to provide an additional constraint as a new regularization within the learning process but also to advocate exploiting the significant performance gains by improving otherwise problematic convergence behavior. Aside from the more diverse parallel studies, implicit deep supervision via shortcut connections began to receive attention from ResNet [26] and DenseNet [27], becoming one of the dominant approaches. FCN [28], PSPNet [29] and UNet++ [30] brought deep supervision into the image segmentation task explicitly in addition to the implicit fashion via skip connections and achieved significant performance gain. Similar approaches further proved effective in SSD [31], and FPN [32] for object detection. The method we proposed here applies the previously reported methods to the video processing regression model for the rPPG task for the first time. Previously, in deep supervision, it was necessary to introduce additional task-specific layers such as classification and segmentation heads. For that reason, the task-specific head with corresponding parameters for training was discarded for actual inference. On contrary, we here demonstrate deeply supervised learning without learning additional parameters by introducing supervised layers that utilize frequency domain information as a newly suggest constraint or regularization.

## 3. Methods

### 3.1. Datasets for rPPG

We use Vision-for-Vitals (V4V) dataset that consists of 179 subjects and 1358 videos in total [33,34]. The V4V dataset contains continuous blood pressure waveform recorded at 1 KHz, frame-aligned HR, and frame-aligned respiratory rate [33,34]. The video was recorded with a frame rate of 25 and a resolution of 1392 × 1040. During the video recording process, 10 activities were conducted with a natural transition from positive emotions to negative emotions. Since the validation and test data lacks ppg ground truth, We used the training dataset only, which includes 555 videos from 92 subjects. The dataset was divided into training, validation, and test of the ratio 60%, 20%, and 20% where subjects were mutually exclusive respectively.

The UBFC-RPPG [35] database comprises 42 videos of 42 subjects captured in a realistic setting with a spatial resolution of 640 × 480 pixels and a frame rate of 30 fps. Each video was for approximately 2 min, during which the subjects played a time-sensitive mathematical game to induce variations in their heart rates. The data were randomly split into 21 subjects for training, 8 for validation, and 13 for testing on a total of 42 subjects for our experiments.

The PURE [7] database includes 60 videos recorded from 10 subjects (8 male and 2 female) in 6 different setups. Each video is 1 min long and has a resolution of 640 × 480 pixels and a frame rate of 30 fps. The subjects were instructed to perform 6 different types of head motions, including steady, talking, slow translation, fast translation, small rotation, and medium rotation. We conducted a holdout split in which subjects 1–8 were for training and subjects 9 and 10 were for testing, ensuring that the subjects were mutually exclusive. For the V4V database [33,34], we conducted experiments under the same conditions as the previous experiment optimizing PhysNet as a lightweight network.

### 3.2. Implementation Details

In this research, we conduct the training and inference phases of our DSE-NN using the PyTorch framework. The experiment runs on an NVIDIA RTX 8000 GPU. For the implementation of the MCC loss function, we utilize the Fast Fourier Transform (FFT) module within the PyTorch library (torch.fft). Moreover, for the evaluation of PPGsignals, we employ the Short-Time Fourier Transform(STFT) module from the SciPy library (scipy.signal.stft).

### 3.3. Spatiotemporal Encoder-Decoder Network

Using the above dataset, we optimized the spatiotemporal network by reducing the number of channels of the PhysNet encoder-decoder model, which reported the best performance among the models proposed by Yu et al. [20]. When we train the PhysNet model several times from scratch, it was observed that the HR prediction performance of the test data varied significantly under the same training conditions. Many previous publications reported small differences in HR prediction error (e.g., less than 0.5%) for comparison in ablation studies but we found that these differences are too little to draw a meaningful conclusion with a statistical significance. Therefore, we examined the statistical significance of the performance improvement in the optimization process by observing not only the average value but also the standard deviation of 10 independent experiments. The PhysNet encoder-decoder model starts with a stem layer with 32 channels, followed by the encoder channel width of 64 (Figure 2).

### 3.4. Deeply Supervised rPPG Network

The visualization of the intermediate layer suggests that the hidden layers are already capable of learning periodic representations of heart pulses. For this reason, we introduce a method for deep supervision of rPPG regression networks. For small training data and relatively shallower networks, deep supervision has been demonstrated to serve as a strong regularization and provide significant performance gains with fast convergence behavior. One concern with the direct pursuit of deeply supervised features at all hidden layers is that this might interfere with the overall network performance; our experimental results indicate that this is not the case. The overall structure of the deeply supervised rPPG network is shown in Figure 1. For each layer of the encoder and decoder, only the temporal information was extracted through spatial average pooling. For the ground truth of deep supervision, PPG label was down-sampled in regard to the temporal dimension of each layer. To mitigate the phase misalignment between prediction and ground truth in deep supervision, We employed the maximum cross-correlation (MCC) [36] in the frequency domain as an objective function:(1)$$\mathrm{MCC}=c_{pr}\times \max\left(\frac{F^{-1}\left\{\mathrm{BPass}\left(F\left\{y\right\}\;\cdot \;\bar{F\{\hat{y}\}}\right)\right\}}{\sigma_{y}\times \sigma_{\hat{y}}}\right)\;$$

In the MCC (Equation (Equation 1)), *y* and $\hat{y}$ represent the signals after mean subtraction, focusing on signal fluctuations. The Fast Fourier Transform (FFT), denoted as *F*, is applied to these signals, converting them from time to frequency domain. The conjugate of *F*, represented as barF, is used during the cross-correlation computation. The signals are zero-padded before FFT to prevent circular correlation. A bandpass filter, indicated by BPass, is applied to isolate the heart rate frequencies within the range of 40 to 180 bpm. The standard deviations of the original signals *y* and $\hat{y}$, denoted as $\sigma_{y}$ and $sigma_{\hat{y}}$ respectively, are used for normalization. Finally, $c_{\mathrm{pr}}$ is a constant scaling factor representing the power ratio within the heart rate frequency range, ensuring that the MCC focuses on relevant frequencies. In our proposed model, the use of MCC loss is crucial for its precision in frequency domain analysis and enhanced signal correlation. By employing the FFT, MCC loss accurately analyzes the spectral components crucial for heart rate detection. Moreover, the conjugate multiplication aspect of MCC loss effectively measures the correlation between the physiological signals and the rPPG signal, ensuring a high degree of accuracy in synchronizing the extracted heart rate with the actual physiological heart rate.

A weight of 0.1 was applied to the auxiliary loss of each intermediate layer compared to the primary loss. This proposed method calculates the spectral loss directly without additional parameters or regression heads, whereas in previous studies, an additional task-specific network needs to be added to train the auxiliary loss. Therefore, the network has the advantage of using the same structure in training and testing except for batch normalization.

### 3.5. Optimization of Network Architecture through Cost Function

In our study, we address the optimization of neural network architecture, particularly in terms of channel numbers. We formulated an objective to minimize a cost function that balances the model’s predictive error and its complexity. Due to significant scale differences between these two components, we apply normalization to align their scales effectively. The cost function is defined as:(2)$$\min_{\alpha }\left(\epsilon \left(\alpha \right)+\lambda P\left(\alpha \right)\right)\;$$

Here, ϵ(α) represents the predictive error of the model, and P(α) denotes the number of parameters, reflecting the model’s complexity with respect to the number of channels of intermediate layers α={1,2,4,…,64}. The coefficient λ is a regularization term that control the trade-off between minimizing the error and the model’s complexity, defined by $\sigma_{\epsilon }$/$\sigma_{P}$ where $\sigma_{\epsilon }$ and $\sigma_{P}$ are the standard deviation of the error and the number of parameters, respectively. Our optimization approach focuses on minimizing this cost function. The optimal channel numbers are those that yield the lowest cost, indicating a balanced architecture in terms of performance and complexity. This methodology enables us to systematically fine-tune the network’s architecture, achieving an effective balance between performance accuracy and model size.

We represent the values of our defined optimization objective function in Figure 3. In Figure 3a, α represents the number of channels in the encoder layers, demonstrating how the cost varies with changes in the number of channels of the encoder layers. Similarly, Figure 3b illustrates the impact of varying the number of channels in the decoder layers on the cost, with α denoting the number of channels in the decoder layer. These visualization aid in understanding the relationship between the number of channels in different layers and the overall model optimization, providing clear insights into how different configurations of α affect the balance of predictive error and model complexity.

### 3.6. Evaluation Metrics

The recovered rPPGs and corresponding ground truth PPGs undergo the same process of filtering, normalization, and spectral transformation using short-time Fourier transformation to derive the HR. Our evaluation of performance relied on the root mean square error (RMSE), mean absolute error (MAE), and Pearson correlation coefficient (PCC). We compared the performance of these metrics across 10 experiments for each dataset and different experimental conditions to identify the best performance.

### 3.7. Experimental Setup

For the short-time Fourier transformation, the RMSE value tends to appear smaller as the time window gets longer because the average is ultimately taken over a longer period of time. However, it is not feasible to use an excessively long time window in applications that require frequent updates of BPM or detailed analysis of time-dependent factors such as HRV, as shorter windows provide better temporal resolution. To ensure a fair evaluation, we used the shortest window of 5 s that has been used in other studies for our evaluation. The sliding step in the within-database case is taken as 0.1 s. A smaller sliding step can help to increase the number of training samples for the within-database case. All reference PPG signals are resampled to be aligned with the video frame rate. We train the proposed DSE-NN for 100 epochs using the Adam optimizer with a weight decay coefficient of 0.1. The initial learning rate is set to 0.0001 and the batch size is set to 6.

## 4. Results

We optimized and evaluated the proposed DSE-NN on public datasets to illustrate its effectiveness. The performance of our proposed method was assessed with and without deep supervision. Additionally, we compared it with existing works evaluated on the V4V dataset, UBFC-rPPG, and PURE.

### 4.1. Optimization of Spatiotemporal Network

We optimized the spatiotemporal network by modulating the channel width of the encoders and the decoders sequentially, evaluating the HR RMSE.

As shown in Figure 3a, the performance of the rPPG network was slightly improved, despite the decreasing channel width of the encoder. However, the performance started to deteriorate when the channel width became 1. These results suggest that the baseline model may be an overfitting model compared to the required complexity when tested in the popular benchmark datasets. The model with a channel width of 16 showed the best performance among all evaluated conditions, which was statistically significant and higher than that of the baseline. Additionally, the decrease in the number of model parameters started to slow down to some extent, and therefore, we conclude that the encoder channel width of 16 was optimal. As a result, the HR RMSE was reduced from 7.703 to 6.833, and the number of parameters of the entire model decreased by 91% from 867 k to 78 k. Figure 3b shows the performance change of the model according to the change in decoder channel width. There was little change in performance despite the extreme reduction of the channel width of decoders, where no statistically significant difference was found with the *p* values, which were larger than 0.05.

### 4.2. Visualization of Features and Learned Filters

For a better understanding of learned features by the spatiotemporal model, we visualized the activations produced on each layer of a trained convolutional network and the learned features computed by individual neurons at every layer. In convolutional networks, filters are applied in a way that reflects the underlying geometry of the input and produces activations in spatially arranged layers for each channel. The activations of neurons in each layer of convnets were visualized in response to the video.

Figure 4 shows a snapshot of the activations of each channel (the output of stem-conv1, encoder1-conv2, encoder2-conv4, encoder3-conv2, decoder-conv1, and decoder-conv2 layers). We used an encoder-decoder architecture as a baseline [20] that includes encoder layers (channel widths of 32, 64, 64, and 64) and decoder layers (channel widths of 16 and 8). For example, the second conv layer output of encoder1 (Figure 4b at a given temporal point has a size of 64 × 56 × 56 (C × W × H), which we depicted as 64 separate 56 × 56 grayscale images. Each of the 32 small image patches contains activations in the same spatial x-y spatial layout as the input data, and the 32 images are simply and arbitrarily tiled into a 6 × 6 grid in row-major order. The activation of the convnets focused on the skin region except for the eyes and lips with the forehead or cheeks accentuated. The deeper the layer, the more information is kept in an abstract geometrical form.

Surprisingly, decoder activation does not learn various spatial patterns across channels, and the learned spatial activations converge to one or two patterns. In this context, we visualized how the waveform changes over time by averaging the activation values for each channel in space. Figure 5 shows the temporal activation pattern of the decoder channel of the trained model while reducing the number of channels from 64, 32, 16, 8, 4, to 2. Regardless of the number of channels, the temporal activation converged into two representations. Based on these results, we conducted an experiment with a model with two decoder channels as the final model selected later.

Figure 6 shows the spectral activations of intermediate layers which were visualized by assessing how the temporal representation changes over time by averaging the activation maps of each layer in space for each channel and transforming the temporal signal into the frequency domain. The periodic pattern in time did not appear dominantly in the initial layer. Although it is not a dominant signal in the initial layer, it still showed a periodicity that matched the heart pulse, and the temporal periodic pattern was strengthened as the layer was deepened. Moreover, transformation into the frequency domain made this phenomenon more conspicuous. These results imply that we can help model learning by direct supervision of the intermediate layers rather than propagating it through backpropagation of the output to learn the heartbeat periodicity.

The learned filters were also visualized for the first convolution layers with a filter size of (1, 5, 5) (Figure 4g). We found that the learned filters tend to capture low-frequency information in space, which is presumably important to aggregate the skin pixel information for tracking the heart pulse in time instead of space. Previous works have demonstrated a skin segmentation loss as an auxiliary objective function; however, our result provides intuitions that we need a more sophisticated method to employ spatial semantics and incorporate these methods into the network architecture, which can be investigated in future work.

### 4.3. Ablation Study

To analyze the influence of each module on the experimental performance, we report the results of the ablation study of the proposed method on a set of three recent PPG datasets (V4V, UBFC-RPPG, and PURE) in Table 1. We conducted a comparison between encoder-decoder models of PhysNet that were trained using either MSE loss or NegPCC (Negative Pearson Correlation Coefficient) loss, which was used as a baseline for our study. The two loss functions showed similar performance in PURE, UBFC-RPPG, and V4V, but as anticipated, the MSE loss resulted in improved MSE metrics while performing poorly in PCC metrics. On the other hand, the NegPCC loss resulted in improved PCC metrics while performing poorly in MSE metrics. The same trend was observed in the optimized lightweight PhysNet model. For example, in the PURE dataset, the MSE loss reduced the RMSE by 50% from 4.379 to 2.227 bpm, but it can be observed that the PCC decreased from 0.779 to 0.537. The lightweight PhysNet model showed similar performance on all datasets compared to the original PhysNet model, despite having 11 times fewer parameters. Since we used MCC loss for deep supervision, we also tested it after training the lightweight PhysNet model with MCC loss instead of MSE loss or NegPCC loss to understand the performance of MCC loss. In this case, MCC loss performed poorly on all metrics on PURE, and it showed intermediate performance between MSE loss and NegPCC loss in UBFC-RPPG and V4V. Based on these results, we used MSE loss for PPG regression and MCC loss for deep supervision.

The use of deep supervision can lead to faster convergence rates due to larger gradient values and variances. Additionally, it can increase robustness in hyperparameter selection, as the direct regression loss used in the early layers can result in quicker convergence and less dependence on extensive hyperparameter tuning. To examine convergence behavior, we compared 10 average training curves between baseline, lightweight models with deep supervision (Figure 7). As expected, the convergence rate was accelerated, and the number of epochs converged to HR RSME lower than 10 bpm and decreased from 19 epochs to 8 epochs.

The DSE-NN showed the best performance in RMSE, MAE, and PCC in all datasets except PCC of UBFC-RPPG. Overall, it can be observed that the PURE dataset shows better performance metrics for rPPG prediction than UBFC-RPPG and V4V. This can be speculated due to the fact that V4V and UBFC-RPPG datasets contain various stimuli that induce facial expression changes or sudden facial movements, such as arithmetic calculations or emotional arousal.

The effect of deep supervision can be directly confirmed by comparing a lightweight network with MSE loss to a DSE-NN that includes only deep supervision added to the former. By deep supervision, the lightweight network improved RMSE from 2.227 to 1.062 bpm and PCC from 0.537 to 0.979. Remarkably, a deeply supervised network using MSE loss instead of PCC loss even showed better PCC values than the lightweight network using PCC loss (0.779) and PhysNet using PCC loss (0.810). This trend was observed not only in the PURE dataset but also in the UBFC-RPPG and V4V datasets. These results indicate that deep supervision contributes to overall performance improvements.

Our proposed approach is not limited to our suggested lightweight encoder-decoder network and can be applied to any neural rPPG regression model that contains periodicity representation. Furthermore, this method is not confined to supervised learning; it can also be used with self-supervised methods to deeply supervise intermediate layers [36].

## 5. Discussion

### 5.1. Impacts of Deep Supervision

The use of deep supervision can lead to faster convergence rates due to larger gradient values and variances. Additionally, it can increase robustness in hyperparameter selection, as the direct regression loss used in the early layers can result in quicker convergence and less dependence on extensive hyperparameter tuning.

To examine convergence behavior, we compared 10 average training curves between baseline, lightweight models with deep supervision (Figure 7). As expected, the convergence rate was accelerated, and the number of epochs converged to HR RSME lower than 10 bpm and decreased from 19 epochs to 8 epochs.

Our proposed model, DSE-NN showed the best performance in RMSE, MAE, and PCC in all datasets except PCC of UBFC-RPPG. Overall, it can be observed that the PURE dataset shows better performance metrics for rPPG prediction than UBFC-RPPG and V4V. This can be speculated due to the fact that V4V and UBFC-RPPG datasets contain various stimuli that induce facial expression changes or sudden facial movements, such as arithmetic calculations or emotional arousal.

### 5.2. Performance on PURE, UBFC-rPPG, V4V

To verify the effectiveness of the DSE-NN, we compared our proposed method with previous works, including traditional methods [15,37,38,39] and methods based on deep learning [10,20,21,22,33,36,40,41,42,43] (Table 2). Compared to the signal processing-based methods (CHROM [18,39], POS [15,18]), deep learning methods (Wang et al. [40], Gideon et al. [36], HR-CNN [22]) showed better performance by large margin in PURE. DSE-NN showed the best performance in PURE except for PCC which the work of Gideon et al. [36] showed a PCC of 0.99 compared with our method with a PPC of 0.98. On the UBFC-RPPG dataset, our proposed method outperformed other deep learning methods except for PulseGAN [41]. However, our method can also be applied to deep learning methods including PulseGAN, and it can be expected to be used as an additional module to improve the performance of existing deep learning models. On the V4V dataset, our method exceeds the performance of both signal processing-based methods (GREEN [38], ICA [37], POS [15]) and deep learning methods (DeepPhys [21], Revanur et al. [33]). Furthermore, as deep supervision methods, our model optimization and visualization methods are generally applicable and can be used as general tools that can be applied to those deep learning methods beyond the network architecture.

### 5.3. Comparison of Computational Complexity

In the comparison of computational complexity across various rPPG models (Table 3), our proposed DSE-NN model shows notable efficiency. DSE-NN model stands out due to its significantly lower number of parameters, 57k, which is an order of magnitude smaller than most competitors, such as DeepPhys [21] and HR-CNN [22], which possess parameters in the millions. This reduction in parameters implies a lightweight model, facilitating faster training times and reduced memory footprint.

Furthermore, DSE-NN model’s FLOPs are calculated to be 44.26 × $10^{9}$ which is substantially lower than that of HR-CNN [22]’s 988.97 × $10^{9}$, suggesting a much less computationally intensive process. This efficiency is pivotal for deployment in real-time applications where computational resources are constrained. Even when compared to models with fewer layers, like Liu et al. [16], DSE-NN maintains a competitive edge in FLOPs without compromising on performance capabilities. In essence, DSE-NN encapsulates the essence of computational frugality while maintaining high accuracy, making it an ideal candidate for real-world rPPG applications that require both efficiency and efficacy.

## 6. Conclusions

This paper presents a deeply supervised efficient network for rPPG measurement from raw facial videos, named DSE-NN. Here, we propose a method to visualize the spatial, temporal, and spectral representation of the spatiotemporal network. DSE-NN optimizes the spatiotemporal network and improves prediction performance. Deep supervision improved the convergence behavior and HR prediction accuracy without additional task-specific trainable parameters with additional performance improvements.

While our proposed model demonstrates promising results in rPPG measurement from raw facial videos, there are areas where further research is needed. A key limitation is the need to validate and enhance the performance of DSE-NN in real-time implementations, particularly on embedded systems. Future work could explore the potential of model lightweighting techniques, such as 8-bit quantization, to enhance real-time applicability without significant loss in performance. Jacob et al. [46] provide insights into efficient quantization methods that could be adapted for DSE-NN. Additionally, applying our deep supervision approach to a variety of other models would be valuable to assess its impact on learning speed and overall performance improvement. The versatility of this approach holds potential not only for advancements in remote photoplethysmography but also for broader applications across various fields in deep learning and signal processing. This cross-disciplinary applicability could lead to significant contributions in diverse areas of research.

## Figures and Tables

**Figure 1 bioengineering-10-01428-f001:**
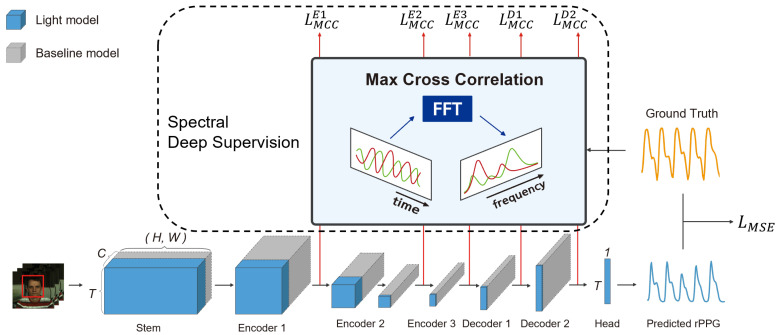
The architecture of DSE-NN. The DSE-NN is an encoder-decoder model, consisting of a stem, encoder, and decoder, along with rPPG head layers. Not only does the final rPPG layer manage loss, but each layer also undergoes spectral deep supervision to accelerate learning and increase accuracy. Spectral deep supervision is implemented through a loss that maximizes the correlation in the frequency domain with the ground truth PPG signal. Using PhysNet as a baseline, the model has been optimized from gray to blue to reduce overfitting, enhance generalization performance, and increase computational efficiency. Smaller boxes denote the lightweight and efficient components of the network. Red arrows illustrate the output of intermediate layers, integral to our deep supervision approach.

**Figure 2 bioengineering-10-01428-f002:**
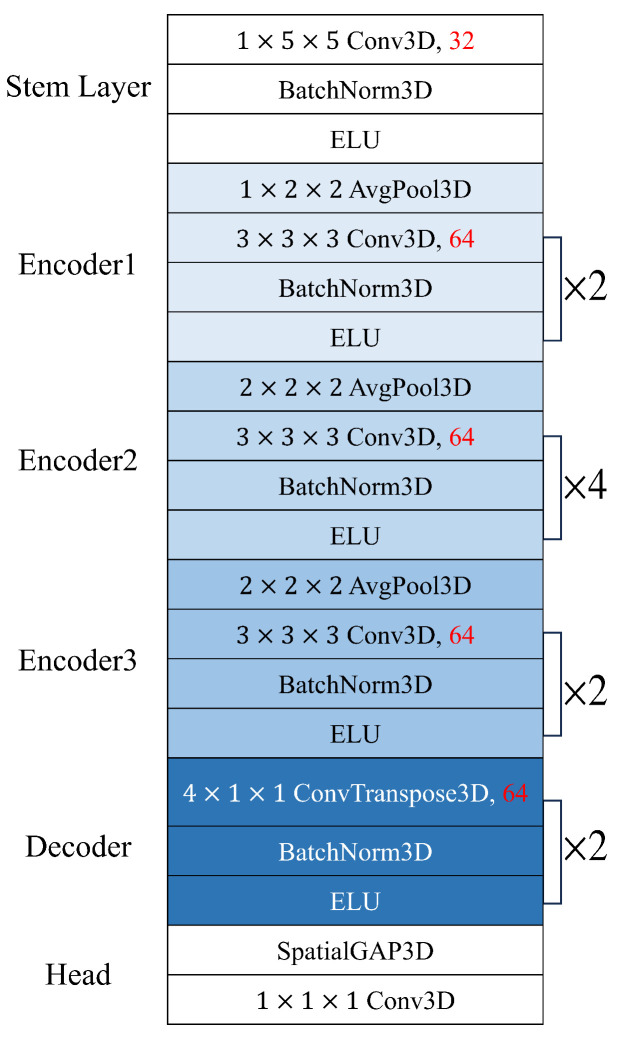
Baseline PhysNet encoder-decoder architecture.

**Figure 3 bioengineering-10-01428-f003:**
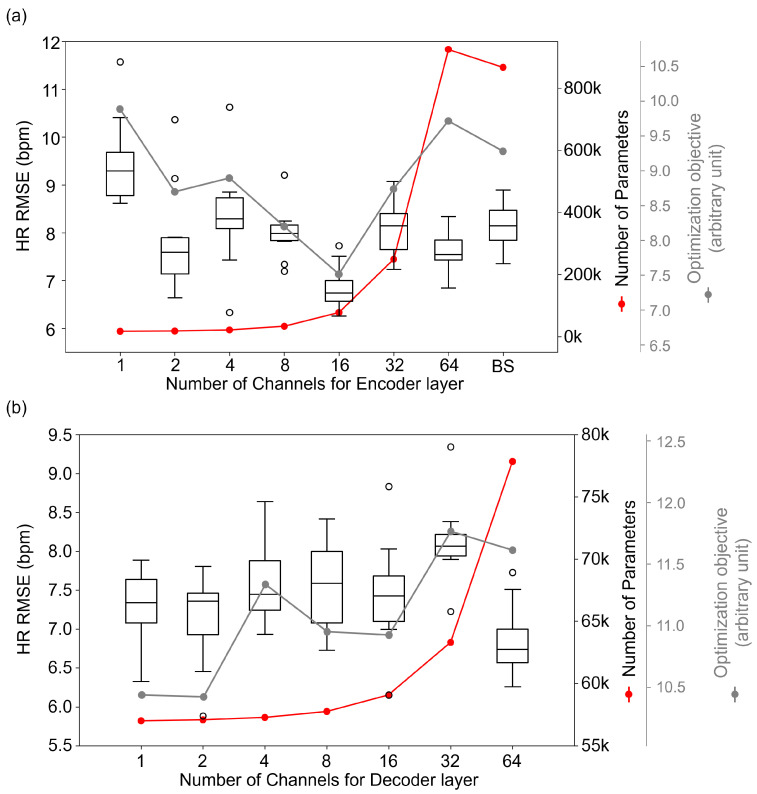
Accuracy according to numbers of channels of encoder (**a**) and decoder layers (**b**). BS stands for Baseline model, PhysNet. Red dots represent the number of parameters corresponding to the number of channels in each layer, and gray dots indicate the optimization objective values for each configuration.

**Figure 4 bioengineering-10-01428-f004:**
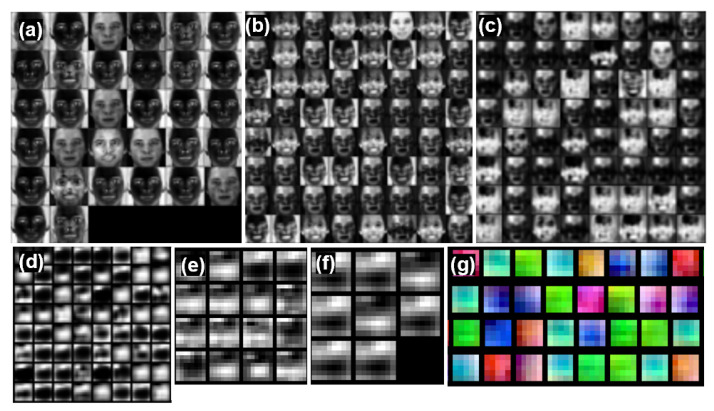
Visualization of activation of (**a**) stem-conv1; (**b**) encoder1-conv2; (**c**) encoder2-conv4; (**d**) encoder3-conv2; (**e**) decoder-conv1; (**f**) decoder-conv2 in gray scale, and learned filters of (**g**) stem-conv1 in RGB. We normalize the signals, thus the y-axes are arbitrary unit.

**Figure 5 bioengineering-10-01428-f005:**
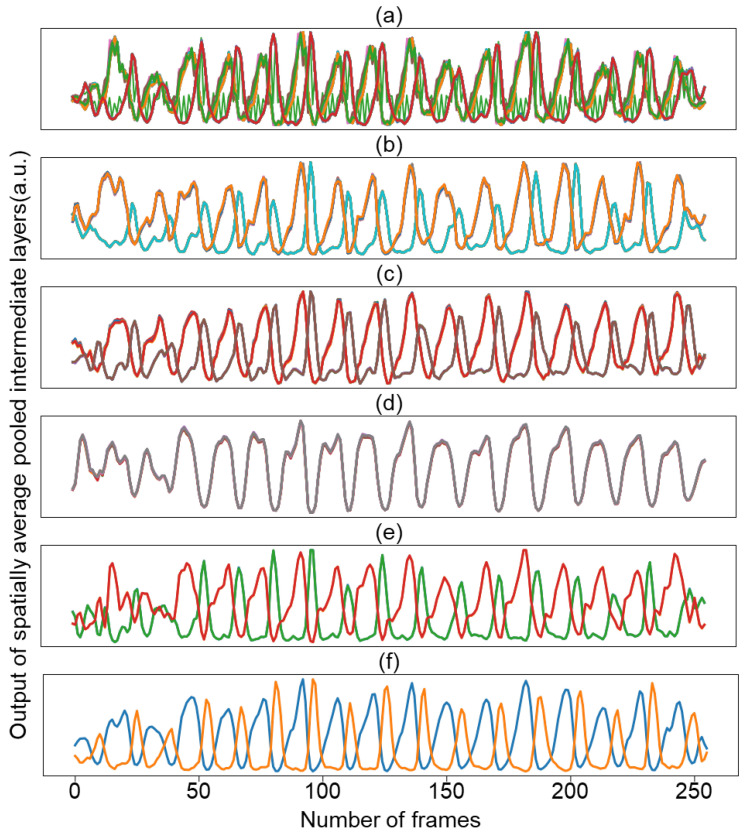
Temporal representation of activations of the decoder-conv2 with the channel number of 64 (**a**); 32 (**b**); 16 (**c**); 8 (**d**); 4 (**e**); and 2 (**f**). Colors in the figure are used for visual differentiation of channels.

**Figure 6 bioengineering-10-01428-f006:**
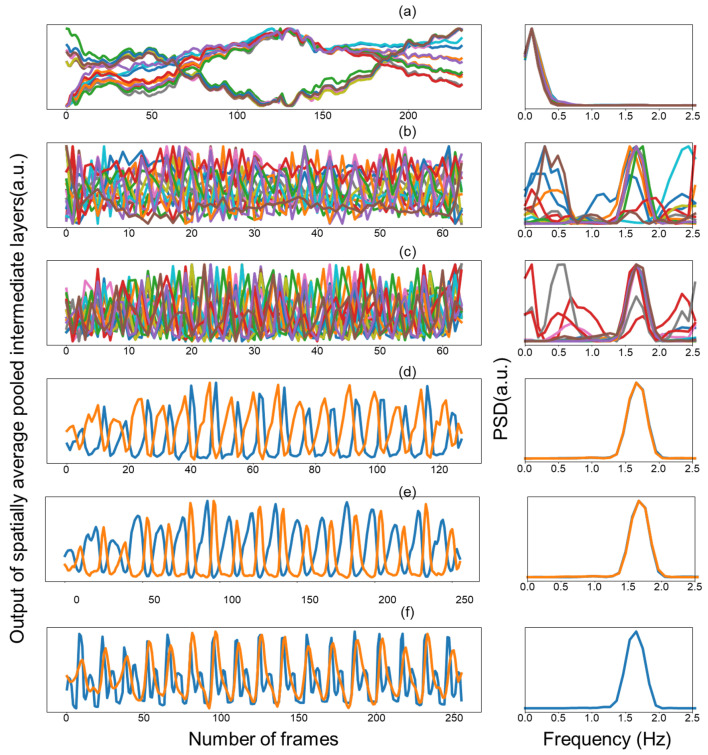
Spatially average pooled output of intermediate layers and frequency domain of each channel (**a**–**e**). Blue line and orange line (**f**) represent ground truth and predicted PPG, respectively. Colors in the figure are used for visual differentiation of channels.

**Figure 7 bioengineering-10-01428-f007:**
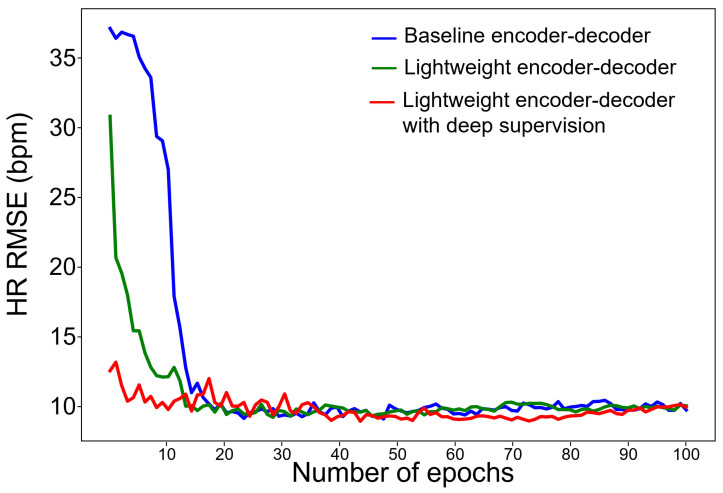
Comparison of convergence rates of each model.

**Table 1 bioengineering-10-01428-t001:** Ablation study of the proposed methods.

	PURE			UBFC-RPPG			V4V		
Method	RMSE	MAE	PCC	RMSE	MAE	PCC	RMSE	MAE	PCC
	(bpm)	(bpm)	(a.u.)	(bpm)	(bpm)	(a.u.)	(bpm)	(bpm)	(a.u.)
Encoder-decoder, MSE	4.833	1.566	0.762	6.693	2.188	0.726	7.441	3.333	0.802
Encoder-decoder, NegPCC	3.750	1.131	0.810	5.558	2.078	**0.887**	6.671	2.830	0.855
Light-weight, MSE	2.227	1.155	0.537	6.324	1.988	0.811	6.765	2.946	0.848
Light-weight, MCC	5.042	1.634	0.141	7.039	2.469	0.817	7.035	2.968	0.858
Light-weight, NegMCC	4.379	1.401	0.779	7.566	2.453	0.841	7.935	3.745	0.844
Light, DS (MCC), MSE	**1.062**	**0.652**	**0.979**	**4.339**	**1.618**	0.764	**6.650**	**2.821**	**0.873**

**Table 2 bioengineering-10-01428-t002:** Results of HR measurement on the PURE (a), UBFC-rPPG (b), and V4V (c) datasets.

(a) PURE			
Method	RMSE (bpm)	MAE (bpm)	PCC (a.u.)
Comas et al. [18], CHROM [39]	2.50	2.07	0.99
Comas et al. [18], POS [15]	10.57	3.14	0.95
Wang et al. [40]	11.81	9.81	0.42
Gideon et al. [36]	2.9	2.3	**0.99**
HR-CNN [22]	2.37	1.84	0.98
Proposed method	**1.06**	**0.65**	0.98
**(b) UBFC-RPPG**			
Method	RMSE (bpm)	MAE (bpm)	PCC (a.u.)
PhysNet [20]	5.10	4.12	0.83
Meta-rPPG [42]	7.42	5.97	0.53
PulseGAN [41]	2.10	**1.19**	**0.98**
Gideon et al. [36]	4.6	3.6	0.95
AND-rPPG [43]	4.75	3.15	0.92
Proposed method	**4.34**	1.62	0.76
**(c) V4V**			
Method	RMSE (bpm)	MAE (bpm)	PCC (a.u.)
GREEN [38]	21.9	15.5	-
ICA [37]	20.6	15.1	-
Comas et al. [18], POS [15]	21.8	15.3	-
DeepPhys [33]	19.7	14.7	-
Revanur et al. [33]	18.8	13.0	-
Proposed method	**6.65**	**2.81**	**0.87**

**Table 3 bioengineering-10-01428-t003:** Comparative analysis of the proposed model with other models, showing differences in input size, number of layers, number of parameters and computational operations.

Name	Input Size	# of Layers	# of Parameters	FLOPs ($\times 10^{9}$)
DeepPhys [21,44]	3 × 150 × 36 × 36 (3 × 256 × 128 × 128)	9	1.46 M	9.62 (207.56)
HR-CNN [22,44]	3 × 300 × 192 × 168 (3 × 256 × 128 × 128)	13	1.87 M	988.97 (428.66)
Liu et al. [16,44]	3 × 150 × 36 × 36 (3 × 256 × 128 × 128)	9	1.45 M	9.61 (207.34)
RhythmNet [44,45]	3 × 10 × 300 × 25 (3 × 256 × 128 × 128)	21	11.42 M	1.70 (95.07)
PhysNet Encoder-Decoder [20]	3 × 256 × 128 × 128	12	866.69 k	438.62
Proposed method	3 × 256 × 128 × 128	12	57.09 k	44.26

## Data Availability

The data presented in this study are available on request from the corresponding author.

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
