# Peer review of "DSE-NN: Deeply Supervised Efficient Neural Network for Real-Time Remote Photoplethysmography"

_bioengineering, 2023, doi:10.3390/bioengineering10121428_

Round 1
Reviewer 1 Report
Comments and Suggestions for Authors
COMMENTS:
• Interesting work. The work considers a deeply supervised, efficient network for measuring rPPG from videos called DSE-NN (notations explained in the work).
• This type of research may be used in medicine and may in some way facilitate diagnosis and further treatment.
• The work is worth publishing.
I SUGGEST additions, e.g. add appropriate comments at the end:
• Bioengineering has its basis in Biomathematics. What mathematical methods are used at work.
• Optimization is mentioned in many places at work. I suggest briefly formulating the relevant optimization issues more precisely.
• What generally available software packages can be used in this type of research. Nowadays, many people use various types of IT tools. In this way, a potential reader/user may be encouraged to implement their ideas.
Reviewer 2 Report
Comments and Suggestions for Authors
(1)The full name of rPPG should be given in abstract.
(2)The cited reference [22] should be put after [14]-[21], in an order. Why you cited it before [14] in introduction section?
(3)In Figure 7, the x-label only has one measure 100?
(4)The computational complexity should be analyzed in the paper.
(5)What is the deficiency of the proposed work? The research plan should be given in the conclusion section.
Comments on the Quality of English LanguageMinor modification
Reviewer 3 Report
Comments and Suggestions for Authors
In this paper, the authors present an rPPG model that can solve the problem of long training and inference time and improve performance. Overall, the paper is well structured, but the following points should be clarified to help readers understand.
1. In the abstract, the performance and convergence rate of the proposed method should be presented briefly and quantitatively compared to other models.
2. An overview of DSE-NN (Deeply Supervised Efficient Neural Network) is needed with reference to Figure 1. It seems reasonable to place Figure 1 at the bottom of the Intro and provide an additional overview of the overall proposed method based on Figure 1.
3. If only ref. is not used for the MCC equation (1), additional explanation is needed about the symbols or constants used in the equation. (ex. Cpr ?, F^(-1): Inverse FFT, F: FFT , barF: conjugate of F, etc., BPass: Bandpass filter)
4. There are table index errors(I ->1, II ->2) and typos. Please correct it appropriately.
Round 2
Reviewer 2 Report
Comments and Suggestions for Authors
The authors have completed the paper revisions according to my comments.